# A Sustainable AI Economy Needs Data Deals That Work for Generators

**Ruoxi Jia**[*]
Virginia Tech

**Luis Oala**[*]
Brickroad

**Wenjie Xiong**
Virginia Tech

**Suqin Ge**
Virginia Tech

**Jiachen T. Wang**
Princeton University

**Feiyang Kang**
Virginia Tech

**Dawn Song**
UC Berkeley

## Abstract

We argue that the machine learning value chain is structurally unsustainable due to an economic data processing inequality: each state in the data cycle from inputs to model weights to synthetic outputs refines technical signal but strips economic equity from data generators. We show, by analyzing seventy-three public data deals, that the majority of value accrues to aggregators, with documented creator royalties rounding to zero and widespread opacity of deal terms. This is not just an economic welfare concern: as data and its derivatives become economic assets, the feedback loop that sustains current learning algorithms is at risk. We identify three structural faults—missing provenance, asymmetric bargaining power, and non-dynamic pricing—as the operational machinery of this inequality. In our analysis, we trace these problems along the machine learning value chain and propose an Equitable Data-Value Exchange (EDVEX) Framework to enable a minimal market that benefits all participants. Finally, we outline research directions where our community can make concrete contributions to data deals and contextualize our position with related and orthogonal viewpoints.

## 1 Introduction

Machine-learning at its core is a data processing chain: data shifts states from inputs to pre-train weights to synthetic outputs. The ascending adoption of AI products has commodified this value chain to a new level and triggered a land-rush for data (Figure 1). The market around this value chain is exploding. Model monetizers that have managed to transform data into a mercantile product see increasing sales. OpenAI alone reported >\$3.5bn revenue in 2024 [1]. However, the distribution of that value is often lopsided. Papers, songs, or lines of code can be scraped, even pushing the envelope of legality [2], and distilled into revenue while data generators often receive little attribution. Data aggregators on the other hand, entities that amass large collections of data from generators, often benefit

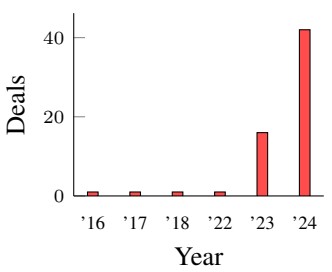

Figure 1: Recent data deals across past calendar years. See Section A for full list.

from the generous terms of service from pre-GPT times. Reddit, for example, banked \$203 million in data licenses until early 2024, yet channeled \$0 to the volunteers who wrote the content [3]. Data and its derivatives have become valuable commodities traded among a small cadre of firms, but the pipeline that transports value from data generators to model monetizers so far appears to be an extraction engine (Figure 2). In our view, the extraction is driven by three mutually reinforcing faults:

---

[*]Equal contribution.

39th Conference on Neural Information Processing Systems (NeurIPS 2025) Position Paper Track.

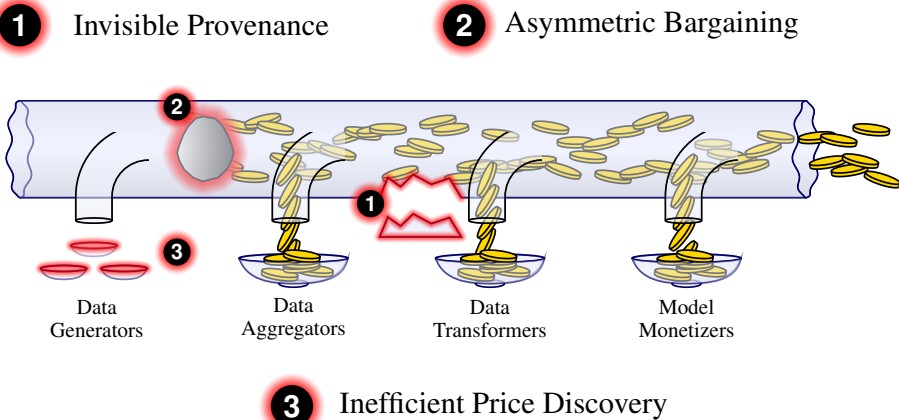

Figure 2: A pipeline symbolizing a piece of the data value chain in machine learning and the structural defects underlying the economic data processing inequality. 1) Data aggregators often strip provenance information of data generators when selling data to companies that transform data into model weights and monetizable products. 2) Model monetizers, which often also transform the data to model weights, enjoy a bargaining advantage as they control much of the current revenue generation. 3) Due to their heterogeneity, data generators in particular are not well equipped to participate in the price discovery of their own data.

**invisible provenance**, whereby pieces of data lose lineage and downstream users cannot audit or route royalties; **asymmetric bargaining power**, in which blanket licences and unilateral API terms let aggregators dictate prices to fragmented contributors; and **inefficient price discovery**, ignoring the dynamic, combinatorial value data accrues in different task contexts.

***Taken together, our position is that the current ML value chain is unsustainable because of an economic data processing inequality that systematically transfers value away from data generators. A sustainable, efficient AI economy needs data deals that work for all market participants including generators.***

If data generators are excluded from the value chain, the supply of high-quality, diverse data will shrink and prices will be set in opaque, concentrated markets. That dynamic stands to harm our own community: researchers, startups, and large labs alike risk facing fewer, less representative datasets—the very fuel current learning algorithms require. Conversely, we hypothesize that open, shared infrastructure facilitating data exchange between all market participants can help to stimluate data flow in an AI economy.

**Contributions.** **(1)** We compile and analyze a collection of 73 publicly disclosed data deals (Section 2). We trace how missing lineage, weak bargaining, and one-shot pricing form a feedback cycle that concentrates capital and cuts out data generators from the value chain. **(2)** In Section 3, we sketch an Equitable Data-Value Exchange Framework that wires task-data matching, dynamic data pricing, and auditable provenance into an efficient marketplace that benefits all participants. **(3)** We surface open problems—from scalable provenance tooling to incentive-compatible marketplaces—that our community can contribute to. **(4)** We weigh our observations and proposal against related (Section 4) and opposing view points (Section 5).

## 2 Economic Data Processing Inequality

Machine learning at its core is a data processing chain. While technical signal is carefully refined along this chain, economic equity is routinely removed from the original data generators—a circumstance we call *economic data processing inequality*. In our analysis we identify three structural mechanisms convolving into this processing inequality: *invisible provenance*, *asymmetric bargaining power*, and *inefficient price discovery*. These are not just concerns of economic welfare. As foundation models become economic actors or agents in their own right and data becomes a primary asset in that system of value creation, the link between data generation and weight harvesting that sustains today's learning algorithm is under strain. Capital concentration and market inefficiency compound with each generation of data derivative, threatening to disenfranchise data generators. Data aggregators enjoy

particularly powerful bargaining position at this time. This also raises the question whether data transformers, the actors refining data into model weights or labels, and model monetizers are getting the best deal compared to an open market. Figure 3 puts the scale and distribution of value in crass contrast: of the $677.3m in reported revenue, creator royalties round to almost zero. Symptomatically, 57 of the 73 found deals do not disclose any revenue publicly. The problem of dark figures appears widespread on both the number of deals transacted and their revenue volumes. A healthier market must work for model monetizers, data aggregators, and data generators alike.

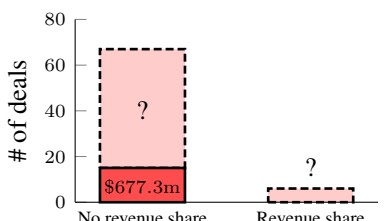

Figure 3: Counts of data deals with and without revenue share information. Left bar (No revenue share): solid segment (15 deals) corresponds to deals where public sources quoted a revenue volume. If ranges are given we conservatively take the floor of that range. Total sum of disclosed revenue volume is $677.3m. The dashed segment (52 deals) indicates additional deals without revenue share and shows the problem of dark figures in this space. Right bar (Revenue share): 6 deals indicate revenue sharing with generators, only one has public information on actual revenue ($2.5k).

**Invisible Provenance.** Once data is copied beyond its point of creation, contextual metadata—license, collection method, consent—often gets lost [4]. Provenance failures are by no means confined to academic benchmarks: they surface in healthcare ([5]), social media ([6]), or open-source code ([7]). As early as 2017, we have cases like the DeepMind–Royal Free breach where the UK Information Commissioner ruled that data generators "were not adequately informed that their data would be used" in deep learning products [8]. These types of disconnects between data's underlying license and its actual use appear to be a structural problem across the board that concerns many ML community datasets today. An analysis on over 1800 datasets has shown that more than 70% have license omissions and datasets with licenses have error rates of over 50% [4]. This causes provenance loss for data generators and legal uncertainty for data transformers and model monetizers. And the implications of this license uncertainty affect the value chain not only during training but also during inference on model weights. Across modalities, models can be shown to regurgitate images [9] and text [2] including content the model provider may not have the license for. In publicly disclosed license deals, the provenance disconnect between data and value generation already shows: a negligible fraction make provisions for revenue sharing with data generators (Section A and Table 2). Provenance gaps cascade through the value chain. The problem compounds once models are fine-tuned on material they themselves have produced [10, 11]. Already models train on a mix of original human content and synthetic content, which creates increasingly complex lineage graphs. Without robust, practical provenance a downstream user cannot audit permissions, enforce attribution, or route royalties, which in turn can chill secondary innovation and disrupt the feedback loop that rewards originators.

**Asymmetric Bargaining Power.** Even when provenance is intact, individual data generators are typically left out of deals between aggregators and model monetizers. Community platforms such as Reddit ($60 million per annum from Google [12]), Stack Overflow (bundled into Gemini [13]), and stock libraries like Shutterstock (multiple eight-figure licences in 2023–2024) all negotiate encompassing deals while offering creators click-wrap terms that grant sweeping reuse rights. Despite monitoring agencies such as the US Federal Trade Commission warning companies that such practices may be deemed deceptive [14], many platforms that host user-generated content (UGC) such as Google, Adobe, Snap, X or Meta have been modifying their terms of service regardless [15]. This power asymmetry also comes to bear in publicly disclosed data deals. More than 40% of known transactions were conducted by OpenAI/Microsoft, Google, or Perplexity on the buying side (Table 1), while aggregators bundle the receipts. These imbalances can snowball. Low marginal inference cost and winner-takes-most network effects channel control surplus to a

|  | Category | Deals |
|---|---|---|
| **Top Types** | News | 26 |
|  | Images | 16 |
|  | Academic | 15 |
|  | UGC | 14 |
| **Top Buyers** | OpenAI | 24 |
|  | Undisclosed | 8 |
|  | Google | 6 |
|  | Perplexity | 3 |
| **Payment** | Amount disclosed | 16 |
|  | Undisclosed | 57 |
|  | Recurring | 3 |
|  | Generator split | 6 |
|  | Litigation | 4 |

Table 1: Aggregated snapshot on types, buyers and payment of 73 publicly disclosed data deals from Table 2.

handful of aggregators and monetizers. Unintuitively, this asymmetry also has the potential to harm the model monetizers who buy the data, because they negotiate with aggregator platforms rather than the generators directly in an open market. Rosen's "superstar" economics [16] suggests—and recent market caps of large AI companies confirm—that the top few firms stand to capture a lion's share of AI rents. Of the 73 transactions in Section A only six mention any revenue-share with contributors. In contrast, several deals, marked "L", are under litigation, signalling that they were hedged under the prospect of court action rather than negotiated at arm's length.

**Inefficient Price Discovery.** Where money does change hands it is often a lump-sum buy-out. News media licenses are illustrative: Associated Press agreed a two-year, flat-fee deal (amount undisclosed) [17]; News Corp settled for roughly $250 million across five years [18]; Axel Springer and Dotdash Meredith followed the same template. To the extent of public disclosure, these contracts do not appear to include royalty escalators for generators tied to usage, retraining, or downstream revenue. All of this plays out against a shifting legal backdrop: ongoing cases such as *Getty Images v. Stability AI*, *NYT v. OpenAI*, and *News/Media Alliance v. Cohere* hint that copyright doctrine, usage rights, and privacy statutes have yet to converge on generative training. Until clarity emerges, originators must choose between costly litigation and acquiescing to not participate in the market. In this sense, static payments are reinforced by the provenance gap. Once an originator has been taken out of the market equation, they have no financial stake anymore to participate in data quality, maintainenance of consent records, or police misuse, while the buyer internalizes much of the upside of any future innovation.

These three faults interact multiplicatively: missing provenance, undercuts bargaining, weak bargaining yields one-shot buy-outs, and buy-outs remove any incentive to invest in provenance. Breaking the cycle and creating an open market that maximizes overall welfare therefore requires technical and institutional interventions that address *all three dimensions at once*.

## 3 Towards an Equitable Data-Value Exchange Framework

The empirical cracks identified in Section 2 prompt the design of a data pipeline ensuring bargaining symmetry, provenance, and efficient pricing. This section outlines the Equitable Data-Value Exchange Framework (EDVEX), a minimal blueprint designed to empower diverse data contributors—especially smaller ones—and align market interests (Figure 5). We detail its potential layers and highlight key open problems, inviting discussion and future research.

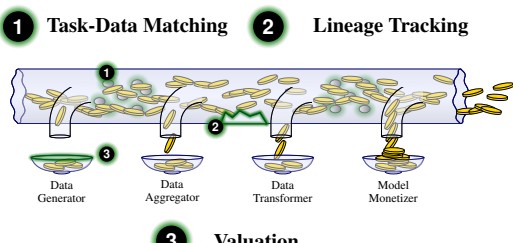

Figure 4: EDVEX patches for a sustainable and efficient machine learning economy.

### 3.1 Technical Primitives

**Task-Data Matching.** A foundational layer in an EDVEX would aim to optimally match *data sources* to specific machine learning tasks so as to maximize defined objectives for the model's effective task fulfillment [19]. In a vast and growing data landscape, especially one encouraging diverse contributors, automated discovery tooling becomes increasingly helpful. Simultaneously, data sellers—particularly smaller ones—often lack the resources or foresight to identify every context where their datasets might be valuable. This layer is foundational because it tackles the inefficiencies and information asymmetries that currently hinder effective data utilization. It ensures that data can be surfaced based on its utility for a task, rather than solely on the marketing capabilities or existing relationships of the data provider, thereby fostering a more level playing field and broader participation.

One might envision this as a sophisticated recommendation system for datasets, navigating the vast universe of potential sources to identify *combinations of sources* predicted to yield the highest improvement for a given target task [20, 21]. Traditional dataset discovery, often reliant on keyword searches, struggles to predict a dataset's true utility for downstream ML tasks [22]. Extensive results on empirical scaling laws [23, 24, 25] demonstrate that performance gains on representative data subsamples and small model sizes can help predict gains from the full corpus and larger model size, enabling us to leverage a *sandbox-first evaluation protocol*. Within this sandbox, candidate

datasets (or their representative subsamples) are subjected to small-scale experiments (e.g., using small-scale proxy models), and the resulting performance changes are extrapolated to estimate their marginal utility for the target model at the desired data scale [26, 20, 27, 28, 29]. Datasets are then ranked by this task-specific utility estimate, with the highest-scoring individual sources or bundles recommended to the developer. The sandbox can be designed to include additional features such as optimizing the mixture of high-ranked datasets, the implementation of which could build upon a wealth of existing and ongoing research employing small-scale experiments to inform training data selection [30, 31, 32, 33]. The layer could also include data wrangling operations [34] (e.g., cleaning, transformation, and format standardization) or data pruning/datapoint selection [35, 36, 37, 38] before performing utility estimation.

Overall, by evaluating and potentially combining disparate data sources based on their collective utility for a task, this layer could facilitate the organic formation of *dynamic, task-optimized data unions*. These adaptive groupings, formed based on specific task requirements rather than pre-defined domains as seen in traditional data collectives [39], could empower contributors by creating value more efficiently, which in turn increases their bargaining power against large aggregators.

---

**Open Problems for Task–Data Matching**

**Data profiling under constraints.** How can we design a profile a data source in ways that capture its potential utility for specific AI tasks—facilitating better data discovery and matching—while preserving data contributor privacy and ensuring that the profile itself does not diminish the incentive for data acquisition by prematurely disclosing excessive value [40, 19]?

**Task profiling for effective matching.** How can AI task descriptions effectively articulate model-specific requirements—such as existing data summary, intended model architecture, whether training is from scratch or based on a pre-trained model—to guide the contribution of high-value, relevant data that demonstrably improves downstream model performance [41]?

**Scalability of the sandbox protocol.** How can the sandbox evaluation protocol (subsampling, lightweight model runs, utility extrapolation) be implemented to scale efficiently to potentially millions of datasets and thousands of tasks without incurring prohibitive compute costs or latency?

**Generalization of utility estimation.** Current scaling laws have mainly focused on certain data modalities, model architectures, and AI tasks. How well do utility estimates derived from sandbox evaluations generalize across different data modalities (tabular, time-series, graph), model architectures, and complex AI tasks (e.g., reinforcement learning)?

**Feedback loops and adaptive data discovery.** How can the discovery system incorporate feedback from actual downstream model performance (after full data acquisition and use) to continuously refine its utility estimation techniques for new tasks [42, 43, 44]?

---

**Lineage Tracking and Auditable Provenance.** To pay data generators according to their contributions, we must identify who provided data and how data is used. Currently, for AI training, such a lineage tracking or provenance mechanism is not widely adopted. Typically, a dataset comes with a license stating the restriction of using the data; however, the license does not help with tracking how the data is actually used. Existing lineage tracking [45, 46] requires manual effort to add the lineage metadata.

We need a framework to properly log the data source and the usage of data, so that the information can later be used for data valuation. Each asset (including dataset, trained models, and intermediate values) should have lineage metadata indicating what and how the source data impacts the asset. In the case where the dataset comprises data from different data creators, the metadata should faithfully record all the data sources [47], including small contributions. One challenge here is to have an encoding scheme that efficiently represents combinations of data sources from potentially millions of data creators. The second challenge is that in the AI pipeline the resulting model is influenced by how data is used in the workflow, e.g., data filtering decisions, feature engineering choices, training configurations (e.g., hyperparameters), architectural selections, and training randomness, making lineage tracking far more complex than traditional data processing pipelines. For example, different filters (or data curation mechanisms in general) lead to different data selection choices. Meanwhile, data practitioners might not want to reveal all the details of the workflow. Hence, the framework should consider what information to include in the metadata to make lineage tracking accurate enough without significant memory and execution time overhead.

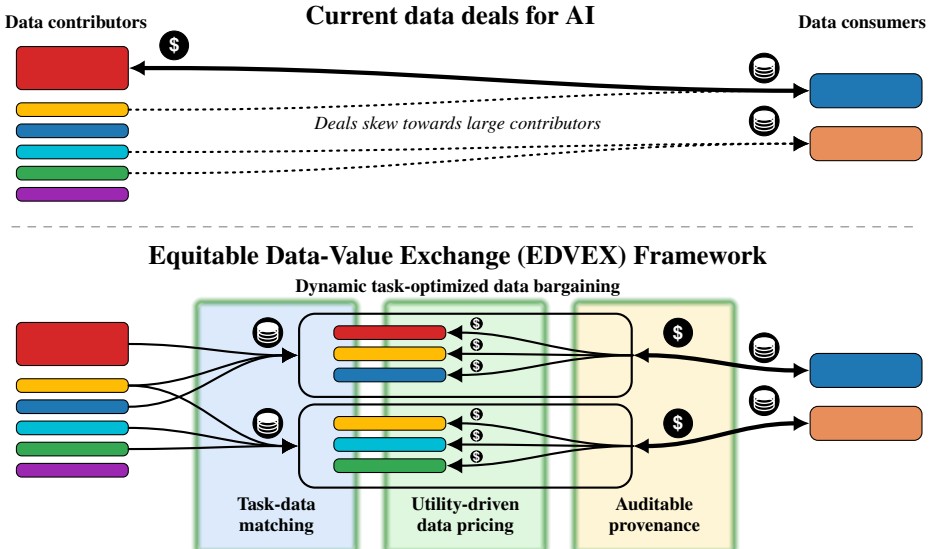

Figure 5: (**Top**): The current landscape of AI data deals is largely dominated by transactions with large-scale content holders (such as major publishers), lacking efficient price discovery mechanisms. Data from smaller players is often scraped *en masse* without compensation or simply overlooked. (**Bottom**): Our envisioned EDVEX Framework features efficient task-data matching, utility-driven data pricing, and auditable provenance to create a more efficient, equitable and transparent ecosystem.

Such a lineage tracking framework should also be designed to be easily deployed by practitioners. Practitioners typically use existing frameworks or APIs, such as PyTorch, for data processing and AI training. To reduce the additional manual efforts for lineage tracking, the lineage tracking framework may be integrated with existing frameworks and logging metadata automatically. Interesting templates for such an architecture include cataloging projects such as Unity [48] and provenance-by-design protocol experiments such as Bittensor [49].

> **Open Problems for Tracking Lineage**
>
> **Information requirements for lineage tracking.** What specific information should be logged to enable effective lineage tracking? How granular should the metadata be regarding individual data creators, transformation processes, and intermediate outputs?
> **Balancing the metadata size and tracking accuracy.** Given the potentially large amount of information needed for accurate lineage tracking, how can we design an efficient encoding mechanism? How should we navigate a trade-off between the metadata size and tracking accuracy?
> **Lower the barrier for tracking lineage.** How can we design the software stack to minimize the manual effort? How can we efficiently ensure complete tracking with robust integrity protection?

**Valuation.** The current paradigm for data acquisition is often characterized by opaque, bilateral negotiations, frequently between large AI developers and data aggregators. This can systematically disadvantage not only smaller contributors but data buyers and may fail to reflect a dataset's true, context-dependent value [50]. This situation can lead to inefficient price discovery and potentially inequitable compensation [51], where significant data sources might be used without recompense or remain unutilized. The "Task-Data Matching" layer, by exploring the formation of dynamic, task-optimized data unions based on preliminary utility estimates, could lay some groundwork for more transparent and equitable valuation approaches. This layer would need to explore mechanisms for two core challenges: establishing a fair market price for an assembled data bundle and ensuring equitable revenue distribution among contributors within that union.

One avenue to explore is leveraging the task-specific utility estimates generated during the discovery phase as a foundation for more transparent price discovery. Instead of relying on bargaining power, the predicted performance uplift (e.g., loss reduction, accuracy gain) offered by a dynamically assembled data bundle could inform its market value. Several approaches could facilitate this. For instance,

AI developers might bid for access to these task-optimized data bundles [52]. The utility estimates from the sandbox evaluation could serve as standard information for bidders, potentially fostering more competitive and fair outcomes. As another example, the price of the bundle could be directly correlated with its estimated contribution to model performance [53]. Besides a one-time upfront payment, AI developers could also offer the union a share of the subsequent value generated by the model developed using their data (e.g., a percentage of revenue). This could align long-term incentives between AI developers and the data contributors. These mechanisms aim to create a more liquid and rational market for data, where price is more closely tied to demonstrable utility.

Once a price for the entire bundle is established, the proceeds can be shared among the contributing data sources. Crucially, for such dynamic data unions to be viable and encourage broad participation, equitable revenue sharing among contributors within the bundle is paramount [54]. If contributors do not perceive the allocation as fair, they may be reluctant to participate [55]. The implementation of equitable revenue sharing can draw inspiration from cooperative game theory concepts like the Shapley values [56], which determine the shares based on a participant's marginal contribution to the overall task-specific utility of the union. The sandbox mentioned above could be designed to provide useful signals to estimate the contribution of individual participants [38, 57].

> ### Open Problems for Valuation
>
> **Efficient and reliable pre-acquisition estimation of data contribution.** What evaluation processes should be conducted within the sandbox, and what specific information about candidate data sources must be made accessible for these evaluations, to enable the reliable and efficient estimation of their individual contributions *before* acquisition?
>
> **Understanding data's influence in complex and iterative AI development workflows.** Modern AI development often involves intricate pipelines with multiple stages, diverse data types, varied training algorithms, and even iterative loops where models are trained on synthetic data generated by earlier model versions. How can we quantify the value contribution of an initial or intermediary dataset as it propagates and transforms through these sophisticated, multi-step processes?
>
> **Contribution to multi-faceted AI evaluation.** How do we design data valuation mechanisms that reward contributions across multi-faceted performance metrics such as fairness and robustness?
>
> **Mitigating "gaming."** Any data valuation system predicated on defined metrics is susceptible to "gaming," where contributors optimize for these metrics, potentially sacrificing genuine data quality [58]. How do we design valuation and market mechanisms that inherently reward genuinely useful data, while actively disincentivizing manipulative behaviors?
>
> **Addressing price erosion for highly substitutable data.** How can valuation and market mechanisms be designed to prevent a "race to the bottom" for data contributions that are abundant and readily substitutable from numerous sources?

### 3.2 Incentives for Implementing EDVEX

The successful realization of an EDVEX hinges on its technical feasibility and, critically, on its ability to create compelling incentives for a diverse range of actors to develop, deploy, and participate in such an ecosystem. This section outlines the primary incentives for key stakeholders.

For **data contributors**, especially smaller players and those with niche datasets often overlooked, EDVEX signals a shift towards equitable participation. The framework's design also promises fair economic returns, determined by the demonstrable utility of their data and transparent revenue-sharing within dynamic data unions. The emergence of a diverse array of data cooperatives and unions demonstrates the growing interest in monetization of their data [59, 39, 60, 61]. Coupled with enhanced discoverability through intelligent matching and auditable lineage, contributors are incentivized by market access and the assurance that their data's value are acknowledged.

This enriched and more accessible data landscape directly benefits **AI developers**. They are incentivized by the prospect of discovering relevant data that can significantly improve model performance, as exemplified by a recent surge of data deals [62]. The task-data matching layer offers a pathway to facilitate and de-risk data acquisitions. The utility-based valuation also offers greater predictability in data expenditure. Moreover, substantial legal and financial risks associated with data misuse provide compelling reasons for framework adoption. For instance, Meta's recent settlements for data privacy

issues—$1.4bn in Texas for biometric data misuse [63] and $725m for the Cambridge Analytica case [64]—dwarf the estimated $677m value of data deals in Figure 3.

EDVEX creates new market opportunities for **platform developers and infrastructure providers** to create and operate the novel services that will define this new data ecosystem—from sophisticated discovery engines and secure sandboxes to reliable lineage trackers and automated valuation tools. Also, by building around EDVEX, these platforms could enhance data exchange efficiency and cultivate trust among data providers, which in turn directly benefits platform adoption.

Ultimately, EDVEX fosters a virtuous cycle: fair compensation and transparency encourage broader data contribution; accessible, high-quality data accelerates AI development; and new infrastructure supports a thriving market.

## 4   Related Work

The call for a more equitable data value chain is not new. Our proposal for an EDVEX builds upon, synthesizes, and extends several lines of existing work spanning conceptual frameworks, organizational models, technical implementations, and specific regulatory mechanisms.

**Conceptual foundations: Data dignity, labor, and dividends**. The philosophical underpinning of EDVEX resonates deeply with concepts like "Data Dignity" and "Data as Labor," notably championed by Jaron Lanier [65] and E. Glen Weyl [66]. These frameworks argue for recognizing the economic value of individual data contributions and advocate for systems where originators are compensated, aligning with EDVEX's core goals of traceable and equitable revenue-sharing. Similarly, proposals for "Data Dividends" (e.g., [67]), distributing profits from data back to contributors. While these ideas provide powerful normative grounding, EDVEX seeks to translate them into an actionable technical and economic protocol.

**Organizational models for collective bargaining**. To address the "asymmetric bargaining power" we identify, various organizational models have been proposed. Data Cooperatives enable members to pool data for collective negotiation and shared benefits (e.g., MIDATA in healthcare [39]). This goal of empowering data subjects through collective action is also central to Delacroix and Lawrence's exploration of bottom-up "data Trusts" [68], and is pursued through collective bargaining mechanisms in what Freedman discusses as data unions [69]. Building on this, this paper discusses mechanisms to facilitate the creation of dynamic data unions. These are specifically designed to empower data contributors—particularly those not part of established collectives—by enabling them to form strategically around the requirements of specific ML tasks, thereby enhancing their capacity to influence ML model development and strengthen their bargaining power.

**Online data marketplaces**. Online data markets aim to enable data deals at scale, but they face significant hurdles regarding data pricing and data quality. Traditional data marketplaces [70, 71, 72, 73, 74, 75], for instance, typically utilize one-time upfront fees, query-based pricing, or subscriptions, which inadequately capture the context-dependent value of data, especially for machine learning applications. These platforms also offer limited information for potential buyers to evaluate dataset suitability, such as limited samples and metadata, making the process of finding suitable, high-quality data inefficient and uncertain. Recent efforts in the Web3 space [76, 77, 78, 49] have focused on creating decentralized data marketplaces, aimed to enhance transparency, introduce token-based compensation, and broaden participation by enabling more, often smaller, players to engage in data transactions. However, they often grapple with the same fundamental challenges of valuation and data quality, which the envisioned EDVEX addresses.

**Regulatory frameworks**. The evolving regulatory landscape increasingly recognizes the need for fairness and transparency in data handling, particularly with the rise of AI. Landmark regulations like the EU's General Data Protection Regulation (GDPR) [79] have established strong protections for personal data, emphasizing consent, data subject rights, and accountability. More recently, the EU Data Act [80] aims to ensure fairness in the allocation of data value in the digital economy, granting users (both individuals and businesses) greater rights to access and share data they co-generate and seeking to rebalance contractual power in data-sharing agreements. While these frameworks establish legal rights and obligations, this paper envisions a set of technical primitives that can help operationalize compliance and foster an ecosystem that embodies their spirit.

# 5  Counter Positions

**Considering alternatives and the cost of inaction.** The proposal for EDVEX stems from the need to address fundamental inefficiencies and inequities in the current data economy. However, it is crucial to consider alternative pathways to these goals and to understand the potential ramifications if these overarching issues remain unaddressed. Several strategies exist to improve the data value chain (see Section 4). However, they often present partial solutions or face limitations when aiming for systemic change. As noted in our discussion of regulatory frameworks (e.g., GDPR, EU Data Act), legal protections are advancing. However, such evolution, alongside current market practices of online data marketplaces (which, as discussed, struggle with robust valuation and discovery), may not sufficiently alter fundamental power imbalances or provide the utility-driven mechanisms EDVEX envisions for fair compensation and optimal data matching. Beyond current regulations, more direct governmental control over data access or value allocation could be pursued. However, this approach risks stifling innovation and may lack the agility to manage dynamic data markets effectively. Web3 initiatives and models like data cooperatives, trusts, and unions (detailed in Section 4) significantly advance transparency, collective bargaining, and contributor empowerment. EDVEX argues for more dynamic, task-optimized collaborations beyond static memberships, recognizing the task-dependent nature of data value. While collaborative ethical pledges by industry can be beneficial, they often lack robust enforcement and may not fully address the systemic representation and compensation issues for smaller data contributors.

Without addressing these fundamental challenges—opaque data valuation, limited access for smaller players, and inadequate compensation—the data economy will become increasingly concentrated, disadvantaging smaller innovators and stifling diverse AI development. These market inefficiencies will perpetuate the systematic undervaluation of data, while unfair practices diminish public trust and risk triggering restrictive regulatory responses [81]. This makes proactive, systemic solutions not just beneficial, but necessary.

**Racing to the bottom.** A significant consideration in designing equitable data deals is the potential for a "race to the bottom" in pricing, particularly for data that may appear abundant or easily substitutable. If many users can provide data suitable for a task, and only a subset is needed, market dynamics might indeed incentivize developers to select the lowest bidders, potentially devaluing the contributions of many. EDVEX's task-data matching layer aims to move beyond simple availability. By focusing on the marginal utility of data for specific tasks, and potentially identifying optimal combinations of diverse sources (as per [20, 31]), the system may value datasets not just on individual merit but on their synergistic contribution. A data set that seems redundant in isolation might offer significant value (e.g., enhancing the representativeness of minority classes) when combined with others, thus resisting pure price-based selection. The concept of "dynamic, task-optimized data unions" is central here. While individual contributors of highly substitutable data might face downward price pressure, unions can provide collective bargaining power. These unions could establish minimum quality thresholds or value propositions for their pooled data, preventing a race to the bottom among their constituents for a given task requiring their specific collective offering. However, for data that is genuinely highly commoditized and where individual contributions offer little unique marginal utility even within optimal bundles, the risk of price depression remains a critical area for future research within the EDVEX framework (see Open Problems for Valuation).

**Data for service.** A pertinent consideration is the prevalent "data-for-service" model [82], where users receive non-monetary value through free access to services. This position paper does not inherently negate this exchange but seeks to bring transparency and fairness, particularly when data's utility extends beyond the immediate service provision. By making data's potential market value explicit through its valuation mechanisms, EDVEX could enable a clearer understanding of the "data" side of the bargain. This could facilitate scenarios where the value of a service is more consciously weighed against the value of data licensed for broader applications, potentially leading to new hybrid models where users are compensated for data uses that transcend their direct service experience.

**Synthetic data.** The increasing use of synthetic data in training AI models, and the emergence of iterative training loops where models generate data for further training ([83, 84, 85, 86, 87]), requires new considerations for a framework like EDVEX. It raises questions about the necessity for valuating human-generated data and the feasibility of tracing real data's value in these complex, recursive pipelines [88]. While synthetic data offers scalability and controllability ([32, 87]), human-generated data will likely remain crucial for grounding models in real-world distributions, nuances, and edge

cases [89]. Synthetic data, especially if generated by models initially trained on other synthetic data, can suffer from a "model collapse" [10, 90]. In domains requiring direct interaction with the physical world—such as robotics ([91, 92]), autonomous driving ([93, 94]), and healthcare ([95, 96]), the need for authentic, real-world data for training, testing, and validation will persist and likely intensify [87]. Synthetic data can augment, but rarely fully replace, data from real-world sensors and interactions in these critical applications [32, 86]. Many valuable datasets are highly specialized, represent niche domains, or fall into the "long-tail" [97, 98]. Synthesizing high-quality, diverse data for these areas without sufficient initial real-world exemplars is extremely challenging [99, 100]. Overall, EDVEX's mechanisms for incentivizing the contribution of real-world datasets remain highly relevant.

It is crucial to also recognize that generating high-quality, diverse, and useful synthetic data is not a trivial or cost-free endeavor [84, 87]. Significant expertise, computational resources, and often sophisticated curation and filtering of initial seed data which may itself be real data are required [86, 88, 101]. EDVEX is agnostic to the origin of the data (real or synthetic) in principle; what matters is its utility, provenance, and the effort involved in its creation and curation. Thus, "data contributors" could indeed be entities (or individuals) who specialize in generating high-value synthetic datasets. We could assess the value of these synthetic contributions just as they would for real data.

# 6 Conclusion

The rapid advancement of artificial intelligence is inextricably linked to the availability and utilization of vast, diverse datasets. Yet, the current paradigms for data exchange are frequently marked by opacity, inefficiency, and an inequitable distribution of value that often disadvantages smaller contributors and hinders optimal data discovery. This position paper has explored EDVEX, a conceptual framework designed to address these foundational challenges. By considering integrated layers for task-data matching and discovery, auditable lineage tracking, and transparent, utility-driven valuation, this paper argues for a community effort to cultivate a data ecosystem that ensures bargaining symmetry, clear provenance, and efficient pricing for all participants.

**Limitations.** Our evidence base relies on publicly disclosed deals and public filings. However, many transactions are private or under NDA, so our dataset likely undercounts and is skewed toward larger, English-language, and U.S.-centric agreements. We therefore report patterns rather than exhaustive statistics. Confidentiality constraints preclude disclosure of non-public deal terms even when known, and we avoid speculation that could compromise sources. Market conditions have been evolving rapidly in the last three years, limiting temporal generalization and our classification necessarily simplifies heterogeneous contracts. While we provide a transparent table, completeness cannot be assumed. As a position paper, we do not present an implementation or pilot. The feasibility and performance of key components remain open research problems.

## Acknolwedgement

Ruoxi Jia and Feiyang Kang acknowledge support from the National Science Foundation through grants IIS-2312794, IIS-2313130, and OAC-2239622. Suqin Ge acknowledges support from the College of Science Dean's Discovery Fund at Virginia Tech. Jiachen T. Wang is supported by Apple's AI/ML PhD Fellowship, Princeton's Yan Huo *94 Graduate Fellowship and Princeton's Gordon Y.S. Wu Fellowship. Luis Oala thanks Bruno Sanguinetti and Freeman Lewin for engaging discussions during the review of the manuscript. The authors thank Alex Izydorczyk, Ithaka S+R and Freeman Lewin for providing resources to cross-reference public deal listings.

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

## A Data Deals - Full Table

| Data Receiver | Data Aggregator | Ref | Date | Type | $ Value | Codes |
|---|---|---|---|---|---|---|
| DeepMind | Moorfields Hospital | [102] | 2016 | Academic | Undisclosed | C |
| DeepMind | NHS | [103] | 2017 | Academic | Undisclosed | C |
| OpenAI | GitHub (Microsoft) | [139] | 2018 | UGC | Undisclosed | L |
| Adobe | Stock Contributors | [104] | 2022 | Images | Undisclosed | C,S |
| Various Licensees | X (formerly Twitter) | [118] | 2023 | UGC | 2.5m/yr | C,R |
| OpenAI | Axel Springer | [114] | 2023 | News | 20m+ | C |
| Apple | Publishers | [105] | 2023 | News | Undisclosed | U |
| ElevenLabs | Voice Actors | [108] | 2023 | UGC | Undisclosed | C,S |
| IBM | NASA | [112] | 2023 | Images | Undisclosed | C |
| LG | Shutterstock | [109] | 2023 | Images | Undisclosed | C |
| Meta | Shutterstock | [110] | 2023 | Images | Undisclosed | C |
| Mubert | Musicians | [111] | 2023 | UGC | Undisclosed | C,S |
| NVIDIA | Getty Images | [113] | 2023 | Images | Undisclosed | C |
| OpenAI | Associated Press | [107] | 2023 | News | Undisclosed | C |
| OpenAI | Shutterstock | [117] | 2023 | Images | Undisclosed | C |
| OpenAI | StackOverflow | [149] | 2023 | UGC | Undisclosed | C |
| Perplexity | Multiple News Publishers | [115] | 2023 | News | Undisclosed | C,S,R |
| Runway | Getty Images | [116] | 2023 | Images | Undisclosed | C |
| Stability AI | AudioSparx | [106] | 2023 | UGC | Undisclosed | C |
| Stability AI | Getty Images | [163] | 2023 | Images | Undisclosed | L |
| Microsoft | Taylor & Francis / Informa | [119] | 2024 | Academic | 10m | C |
| Undisclosed | HarperCollins | [127] | 2024 | Academic | 2.5k/book | C,S |
| Undisclosed | Reuters | [120] | 2024 | News | 22m | C |
| Amazon | Shutterstock | [129] | 2024 | Images | 25-50m | C |
| Apple | Shutterstock | [130] | 2024 | Images | 25-50m | C |
| Google | Shutterstock | [131] | 2024 | Images | 25-50m | C |
| OpenAI | Shutterstock | [132] | 2024 | Images | 25-50m | C |
| OpenAI | News Corp | [142] | 2024 | News | 250m/5yr | C |
| Perplexity | Yelp | [145] | 2024 | UGC | 25m | C |
| Large Tech Company | Wiley | [159] | 2024 | Academic | 44m | C |
| Google | Reddit | [126] | 2024 | UGC | 60m/yr | C |
| Undisclosed | Taylor & Francis / Informa | [151] | 2024 | Academic | 65m | C |
| Undisclosed | Freepik | [128] | 2024 | Images | 6m | C |
| Undisclosed | Tempus | [171] | 2024 | Health | 72.8m | C,R |
| Google | StackOverflow | [125] | 2024 | UGC | Undisclosed | C |
| Meta | Reuters | [134] | 2024 | News | Undisclosed | C,U |
| Midjourney | Tumblr (Automattic) | [153] | 2024 | UGC | Undisclosed | C |
| Midjourney | Wordpress | [154] | 2024 | UGC | Undisclosed | C |
| Musical AI | Symphonic Distribution | [150] | 2024 | Audio | Undisclosed | C |
| NVIDIA | Shutterstock | [148] | 2024 | Images | Undisclosed | C |
| OpenAI | Dotdash Meredith | [124] | 2024 | News | Undisclosed | C |
| OpenAI | TIME | [152] | 2024 | News | Undisclosed | C |
| OpenAI | NYT | [135] | 2024 | News | Undisclosed | L |
| OpenAI | Reddit | [138] | 2024 | UGC | Undisclosed | C |
| OpenAI | Tumblr (Automattic) | [155] | 2024 | UGC | Undisclosed | C |
| OpenAI | Vox Media | [157] | 2024 | News | Undisclosed | C |
| OpenAI | Wordpress | [156] | 2024 | UGC | Undisclosed | C |
| OpenAI | Le Monde | [133] | 2024 | News | Undisclosed | C |
| OpenAI | Prisa Media | [140] | 2024 | News | Undisclosed | C |
| OpenAI | Financial Times | [136] | 2024 | News | Undisclosed | C |
| OpenAI | The Atlantic | [121] | 2024 | News | Undisclosed | C |
| OpenAI | Condé Nast | [123] | 2024 | News | Undisclosed | C |
| OpenAI | Axios | [168] | 2024 | News | Undisclosed | C |
| OpenAI | The Guardian | [165] | 2024 | News | Undisclosed | C |
| OpenAI | Schibsted | [141] | 2024 | News | Undisclosed | C |
| OpenAI | Future plc | [137] | 2024 | News | Undisclosed | C |
| OpenAI | Hearst Magazines | [143] | 2024 | News | Undisclosed | C |
| Potato | Wiley | [158] | 2024 | Academic | Undisclosed | C |
| ProRata AI | Multiple (500+) News Publishers | [146] | 2024 | News | Undisclosed | C,S |
| Undisclosed | Oxford University Press | [144] | 2024 | Academic | Undisclosed | U |
| Undisclosed | Cambridge University Press | [122] | 2024 | Academic | Undisclosed | U |
| Undisclosed | Sage | [147] | 2024 | Academic | Undisclosed | U |
| Amazon | New York Times | [161] | 2025 | News | Undisclosed | C |
| AWS | Wiley | [172] | 2025 | Academic | Undisclosed | C |
| Cohere | News/Media Alliance | [167] | 2025 | News | Undisclosed | L |
| Google | Associated Press | [164] | 2025 | News | Undisclosed | C |
| Mistral AI | Agence France-Presse | [166] | 2025 | News | Undisclosed | C |
| OpenEvidence | NEJM Group | [169] | 2025 | Academic | Undisclosed | C |
| Perplexity | Wiley | [173] | 2025 | Academic | Undisclosed | C |
| Pinterest | Pinterest Users | [170] | 2025 | Images | Undisclosed | C |
| ProRata | AAAS | [54] | 2025 | Academic | Undisclosed | C |
| Undisclosed | De Gruyter Brill | [162] | 2025 | Academic | Undisclosed | U |
| Undisclosed | DataSeeds AI (Zedge) | [174] | 2025 | Images | Undisclosed | C |

Table 2: Joint table of data deals, in chronological order. Codes: C - deals confirmed by public sources, U - unclear deal status, L - litigation, S - revenue share, R - recurring payment. An interactive version of the table is available at `https://research.brickroad.network/neurips20 25-data-deals`

# Data Deal References

[102] Google deepmind and moorfields eye hospital partnership, 2016. Medical AI partnership. `https://www.moorfields.nhs.uk/research/google-deepmind`.

[103] Royal free london nhs foundation trust and deepmind data sharing agreement, 2017. `https://www.royalfree.nhs.uk/news/statement-re-deepmind-data-processing-agreement-during-testing-phase-streams-app`.

[104] Adobe firefly trained on adobe stock contributors' images, 2023. Adobe newsroom, Firefly FAQ. See `https://www.adobe.com/sensei/generative-ai/firefly.html`.

[105] Apple in talks to pay publishers up to $50 million for ai training, 2023. Reuters report. `https://www.reuters.com/technology/apple-explores-ai-deals-with-news-publishers-new-york-times-2023-12-22/`.

[106] Audiosparx signs licensing deal with stability ai, 2023. Accessed: 2024-07-03. See `https://www.musicbusinessworldwide.com/stability-ai-launches-text-to-music-generator-trained-on-licensed-content-via-a-partnership-with-music-library-audiosparx/`.

[107] Chatgpt-maker openai signs deal with ap to license news stories, 2023. `https://apnews.com/article/openai-chatgpt-associated-press-ap-f86f84c5bcc2f3b98074b38521f5f75a`.

[108] Elevenlabs voice marketplace for ai licensing, 2023. ElevenLabs marketplace. `https://elevenlabs.io/voice-data-partnerships`.

[109] Lg partners with shutterstock to advance ai for better life, 2023. `https://www.shutterstock.com/blog/lg-partners-with-shutterstock-to-advance-ai-for-better-life`.

[110] Meta licenses shutterstock music and images for ai training, 2023. Bloomberg Law, Meta MusicGen, Shutterstock press. `https://www.prnewswire.com/news-releases/shutterstock-expands-long-standing-relationship-with-meta-301719769.html`.

[111] Mubert licenses music from musicians for ai generation, 2023. Mubert docs. `https://musically.com/2023/07/12/ai-music-startup-mubert-reaches-100m-tracks-milestone/`.

[112] Nasa and ibm release open-source geospatial ai foundation model, 2023. `https://www.earthdata.nasa.gov/news/nasa-ibm-collaborate-apply-ai-earth-science-data`.

[113] Nvidia announces generative ai collaboration with getty images, 2023. `https://blogs.nvidia.com/blog/generative-ai-getty-images/`.

[114] Openai and axel springer sign content licensing deal, 2023. `https://www.reuters.com/business/media-telecom/global-news-publisher-axel-springer-partners-with-openai-landmark-deal-2023-12-13/`.

[115] Perplexity and publishers agree revenue-sharing model, 2023. `https://pressgazette.co.uk/news/perplexity-publishers-revenue-sharing/`.

[116] Runway strikes getty images deal ahead of gen-2 release, 2023. Accessed: 2024-07-03. See `https://runwayml.com/news/runway-partners-with-getty-images`.

[117] Shutterstock expands ai training data deal with openai, 2023. `https://www.investopedia.com/shutterstock-expands-deal-with-openai-shares-rise-7559349`.

[118] Twitter paid enterprise api access pricing revealed, 2023. `https://mashable.com/article/twitter-elon-musk-paid-enterprise-api-access-pricing`.

[119] Academic publisher strikes ai data deal with microsoft, 2024. Microsoft-Taylor & Francis deal. `https://theconversation.com/an-academic-publisher-has-struck-an-ai-data-deal-with-microsoft-without-their-authors-knowledge-235203`.

[120] Ai data licensing deals (magis), 2024. Reuters deal reference. `https://magis.substack.com/p/ai-data-licensing-deals`.

[121] The atlantic announces content partnership with openai, 2024. `https://www.theatlantic.com/press-releases/archive/2024/05/atlantic-product-content-partnership-openai/678529/`.

[122] Cambridge university press ai licensing opt-in, 2024. Cambridge AI licensing. `https://www.thebookseller.com/news/cambridge-university-press--assessment-writes-to-20k-authors-for-ai-licensing-opt-in`.

[123] Condé nast and openai partnership, 2024. `https://www.condenast.com/news/conde-nast-openai-partnership`.

[124] Dotdash meredith signs multi-year content deal with openai, 2024. Axios, May 29 2024. `https://www.reuters.com/markets/deals/investopedia-owner-dotdash-meridith-signs-content-license-deal-with-openai-2024-05-07/`.

[125] Google and stack overflow announce ai data partnership, 2024. `https://www.wired.com/story/google-deal-stackoverflow-ai-giants-pay-for-data/`.

[126] Google signs $60m reddit data licensing deal for ai training, 2024. `https://www.cbsnews.com/news/google-reddit-60-million-deal-ai-training/`.

[127] Harpercollins' ai deal will pay authors $2,500 per book, 2024. `https://www.404media.co/harpercollins-ai-deal/`.

[128] Inside big tech's underground race to buy ai training data, 2024. Freepik deal reference. `https://www.reuters.com/technology/inside-big-techs-underground-race-buy-ai-training-data-2024-04-05/`.

[129] Inside big tech's underground race to buy ai training data (amazon), 2024. Amazon-Shutterstock deal. `https://www.reuters.com/technology/inside-big-techs-underground-race-buy-ai-training-data-2024-04-05/`.

[130] Inside big tech's underground race to buy ai training data (apple), 2024. Apple-Shutterstock deal. `https://www.reuters.com/technology/inside-big-techs-underground-race-buy-ai-training_data-2024-04-05/`.

[131] Inside big tech's underground race to buy ai training data (google), 2024. Google-Shutterstock deal. `https://www.reuters.com/technology/inside-big-techs-underground-race-buy-ai-training-data-2024-04-05/`.

[132] Inside big tech's underground race to buy ai training data (openai), 2024. OpenAI-Shutterstock deal. `https://www.reuters.com/technology/inside-big-techs-underground-race-buy-ai-training-data-2024-04-05/`.

[133] Le monde signs ai partnership with openai, 2024. https://www.lemonde.fr/en/about-us/article/2024/03/13/le-monde-signs-artificial-intelligence-partnership-agreement-with-open-ai_6615418_115.html.

[134] Meta signs deal with reuters to bring ai news to its platforms, 2024. https://www.axios.com/2024/10/25/meta-reuters-ai-news-facebook-instagram.

[135] Nyt v. openai: The times's about-face, 2024. Litigation reference. https://harvardlawreview.org/blog/2024/04/nyt-v-openai-the-timess-about-face/.

[136] Openai and financial times announce partnership, 2024. https://aboutus.ft.com/press_release/openai.

[137] Openai and future partner on specialist content, 2024. https://openai.com/index/openai-and-future-partner-on-specialist-content/.

[138] Openai and reddit announce content partnership, 2024. Accessed: 2024-07-03. See https://www.theverge.com/2024/5/16/24158529/reddit-openai-chatgpt-api-access-advertising.

[139] Openai faces early appeal in first ai copyright suit from coders, 2024. Bloomberg. https://www.datacenterknowledge.com/regulations/microsoft-github-openai-hit-with-code-copyright-lawsuit.

[140] Openai partners with prisa media, 2024. https://openai.com/index/global-news-partnerships-le-monde-and-prisa-media/.

[141] Openai partners with schibsted media group, 2024. https://openai.com/index/openai-partners-with-schibsted-media-group/.

[142] Openai to start using news content from news corp. as part of a multiyear deal, 2024. https://apnews.com/article/openai-news-corp-a49144d381796df5729c746f52fbef19.

[143] Openai will bring hearst content to chatgpt, 2024. https://venturebeat.com/ai/openai-will-bring-cosmopolitan-publisher-hearsts-content-to-chatgpt.

[144] Oxford university press actively working with ai companies, 2024. LLM-Oxford deal. https://www.insidehighered.com/news/quick-takes/2024/08/05/oxford-university-press-actively-working-ai-companies.

[145] Perplexity chatbot yelp suggestions data ai, 2024. Perplexity-Yelp deal. https://www.theverge.com/2024/3/12/24098728/perplexity-chatbot-yelp-suggestions-data-ai.

[146] Prorata partners with dmg media, guardian, sky news and others, 2024. https://pressgazette.co.uk/platforms/prorata-ai-dmg-media-guardian-sky-news/.

[147] Sage confirms talks to license content to ai firms, 2024. LLMs-Sage deal. https://www.thebookseller.com/news/sage-confirms-it-is-in-talks-to-license-content-to-ai-firms.

[148] Shutterstock integrates generative ai across stock content service, 2024. https://www.businessinsider.com/shutterstock-integrated-gen-ai-stock-digital-photo-video-content-service-2024-12.

[149] Stack overflow signs data licensing agreement with openai, 2024. Accessed: 2024-07-03. See https://techcrunch.com/2024/05/06/stack-overflow-signs-deal-with-openai-to-supply-data-to-its-models/.

[150] Symphonic opens its catalogue for licensed ai training, 2024. https://musically.com/2024/08/21/symphonic-opens-its-catalogue-up-for-licensed-ai-training/.

[151] Taylor & francis ai deal sets worrying precedent, 2024. Undisclosed-Taylor & Francis deal. https://www.insidehighered.com/news/faculty-issues/research/2024/07/29/taylor-francis-ai-deal-sets-worrying-precedent.

[152] Time magazine strikes licensing deal with openai and perplexity, 2024. TIME press release, Jun 26 2024. https://www.reuters.com/technology/artificial-intelligence/openai-signs-multi-year-content-deal-with-time-magazine-2024-06-27/.

[153] Tumblr and wordpress to sell users' data to train ai tools, 2024. Midjourney-Tumblr deal. https://www.404media.co/tumblr-and-wordpress-to-sell-users-data-to-train-ai-tools/.

[154] Tumblr and wordpress to sell users' data to train ai tools (midjourney), 2024. Midjourney-Wordpress deal. https://www.404media.co/tumblr-and-wordpress-to-sell-users-data-to-train-ai-tools/.

[155] Tumblr and wordpress to sell users' data to train ai tools (openai), 2024. OpenAI-Tumblr deal. https://www.404media.co/tumblr-and-wordpress-to-sell-users-data-to-train-ai-tools/.

[156] Tumblr and wordpress to sell users' data to train ai tools (openai), 2024. OpenAI-Wordpress deal. https://www.404media.co/tumblr-and-wordpress-to-sell-users-data-to-train-ai-tools/.

[157] Vox media announces partnership with openai, 2024. Accessed: 2024-07-03. See https://www.theverge.com/2024/5/29/24167072/openai-content-copyright-vox-media-the-atlantic.

[158] Wiley creates ai partnership program, 2024. Potato-Wiley deal. https://www.publishersweekly.com/pw/by-topic/industry-news/industry-deals/article/96248-wiley-creates-ai-partnership-program.html.

[159] Wiley expects to make $44 million from ai partnership, 2024. Wiley deal reference. https://www.booksandpublishing.com.au/articles/2024/09/04/258068/wiley-expects-to-make-us44-million-from-ai-partnership-authors-unable-to-opt-out/.

[160] Aaas and prorata content licensing pilot, 2025. AAAS-ProRata deal. https://www.eurekalert.org/news-releases/1071967.

[161] Amazon signs ai licensing deal with the new york times, 2025. https://www.nytimes.com/2025/05/29/business/media/new-york-times-amazon-ai-licensing.html.

[162] De gruyter brill ai for authors, 2025. De Gruyter Brill AI deal. https://degruyterbrill.com/en/ai-for-authors/#:~:text=Why%20does%20De%20Gruyter%20Brill%20want%20to%20enter%20agreements%20with%20generative%20AI%20providers.

[163] Getty images sues stability ai over copyright infringement, 2025. https://www.dreyfus.fr/en/2025/02/19/getty-images-us-inc-and-others-v-stability-ai-ltd-2025-ewhc-38-ch-an-interesting-case-in-ai-and-intellectual-property-law/.

[164] Google signs deal with ap to deliver up-to-date news through gemini, 2025. `https://www.ap.org/media-center/ap-in-the-news/2025/google-signs-deal-with-ap-to-deliver-up-to-date-news-through-its-gemini-ai-chatbot/`.

[165] Guardian media group announces strategic partnership with openai, 2025. `https://www.theguardian.com/gnm-press-office/2025/feb/14/guardian-media-group-announces-strategic-partnership-with-openai`.

[166] Mistral signs deal with afp, 2025. `https://ca.finance.yahoo.com/news/mistral-signs-deal-afp-offer-095158286.html`.

[167] News/media alliance announces industry lawsuit, 2025. Litigation over news data. `https://www.newsmediaalliance.org/news-media-alliance-announces-industry-lawsuit/`.

[168] Openai will fund axios local newsrooms, 2025. `https://www.axios.com/2025/01/15/open-ai-axios-local-newsrooms-funding-deal`.

[169] Openevidence and nejm group announce partnership, 2025. OpenEvidence-NEJM deal. `https://www.openevidence.com/announcements/openevidence-and-nejm`.

[170] Pinterest privacy policy, 2025. Pinterest user data policy. `https://policy.pinterest.com/en/privacy-policy`.

[171] Tempus reports second quarter 2025 results; $3.7m in 1h 2024, 2025. `https://www.tempus.com/news/tempus-reports-second-quarter-2025-results/?srsltid=AfmBOoqkGt9igPgNkP7xs-nB9mQHrrmXvWg_oQgfFKvzYnaVLSfzeYjK`.

[172] Wiley announces collaboration with amazon web services, 2025. AWS-Wiley deal. `https://newsroom.wiley.com/press-releases/press-release-details/2025/Wiley-Announces-Collaboration-With-Amazon-Web-Services-AWS-to-Integrate-Scientific-Content-Into-Life-Sciences-AI-Agents/default.aspx`.

[173] Wiley expects to make $44 million from ai partnership (perplexity), 2025. Perplexity-Wiley deal. `https://www.booksandpublishing.com.au/articles/2024/09/04/258068/wiley-expects-to-make-us44-million-from-ai-partnership-authors-unable-to-opt-out/`.

[174] Zedge launches dataseeds ai content marketplace for ai training, 2025. `https://www.stocktitan.net/news/ZDGE/zedge-launches-data-seeds-ai-a-content-marketplace-for-ai-training-yzg8lvti96ec.html`.

