# OpenReview forum: "A Sustainable AI Economy Needs Data Deals That Work for Generators"
_NeurIPS.cc/2025/Position_Paper_Track — NeurIPS 2025 Position Paper Track_

### Official Review · Reviewer_FK6j · 2025-08-01

**Significance:** 4
**Presentation:** 3
**Rating:** 7
**Confidence:** 4

**Summary:**

This position paper argues that the current machine learning value chain is structurally unsustainable due to "economic data processing inequality" that systematically transfers value away from data generators to aggregators and model monetizers. Through analysis of 51 publicly disclosed data deals totaling an estimated $1.75 billion, the authors demonstrate that creator royalties effectively round to zero while aggregators capture the vast majority of value. They identify three interconnected structural faults: invisible provenance (loss of data lineage and licensing information), asymmetric bargaining power (individual creators vs. large aggregators), and inefficient price discovery (static lump-sum payments that ignore dynamic data value). To address these issues, they propose an Equitable Data-Value Exchange (EDVEX) Framework featuring task-data matching, auditable lineage tracking, and utility-driven valuation mechanisms. The framework aims to create dynamic data unions that enhance creators' bargaining power while providing better data discovery for model developers.

**Strengths:**

The paper excels in combining rigorous empirical analysis with comprehensive theoretical framework development. The analysis of 51 real data deals provides concrete evidence for abstract economic arguments, while specific examples like Reddit's licensing arrangements make the inequality tangible and compelling. The identification of three interconnected structural faults creates a coherent diagnosis that explains why current approaches systematically disadvantage data generators. The proposed EDVEX framework demonstrates sophisticated thinking about market design, incorporating insights from economics, computer science, and organizational theory. The technical primitives (task-data matching, lineage tracking, utility-driven valuation) are well-motivated and address specific market failures identified in the analysis. The paper successfully bridges multiple disciplines, making contributions to our understanding of both technical ML systems and their economic foundations.

**Weaknesses:**

While the paper provides a compelling diagnosis and comprehensive framework, several limitations affect its practical impact. The analysis necessarily relies on publicly disclosed deals, which may not be representative of the broader (largely private) data marketplace, potentially biasing conclusions about industry practices. The EDVEX framework, while conceptually sound, remains quite abstract with limited discussion of implementation challenges, transition costs, or adoption incentives for existing market participants. Some proposed technical solutions, particularly Shapley value-based revenue sharing, face known scalability and computational challenges that receive insufficient attention. The paper also underestimates potential resistance from current market incumbents who benefit from existing asymmetries, and provides limited analysis of how to overcome these barriers. Additionally, while the framework addresses multiple technical challenges through "open problems," the complexity of coordinating solutions across all components simultaneously may make implementation more difficult than presented.

**Questions:**

1. The paper mentions Shapley values for revenue sharing but doesn't address computational complexity. How would EDVEX handle scenarios with millions of data contributors and complex data lineages?
2. Given the entrenched interests of current market incumbents, what specific strategies would you recommend for transitioning from current practices to EDVEX? How might early adoption be incentivized?
3. How would EDVEX prevent the formation of new oligopolies or market concentration among platform providers who operate the framework infrastructure?

**Alternative Position:**

Yes, and alternative positions are well-considered and addressed by the argument

**Author Identification:**

No.

**Context:**

4

**Discussion:**

4

**Ethics:**

["NO or VERY MINOR ethics concerns only"]

**Position:**

Yes, the paper argues for or against a position related to machine learning.

**Support:**

4

**Thoroughness:**

4

---

### Official Review · Reviewer_nioU · 2025-08-08

**Significance:** 3
**Presentation:** 4
**Rating:** 7
**Confidence:** 2

**Summary:**

The authors highlight biases in the current machine learning ecosystem: while data aggregators, transforms, and model monetizers are compensated for their work that builds on top of data generators, data generators themselves are poorly compensated (if compensated at at all). A long term future for machine learning requires fixing this inequity to ensure that data generator continue to be incentivized for long term data generation. The authors highlight that the current faults lie with missing provenance, asymmetric bargaining power, and non-dynamic pricing. The authors propose an Equitable Data-Value Exchange Framework to correct these issues and highlight open questions in this space.

**Strengths:**

- The authors clearly articulate the issues with the current machine learning pipeline. As someone who does not necessarily work in these areas, the discussion was incredibly eye opening and thoroughly informative. I truly enjoyed reading this piece
- The authors successfully acknowledge opposing viewpoints without dismissing them outrightly
- I have not seen an actual analysis of public data in a position paper, and I think it was really smart of the authors to tie in public data sources in this argument

**Weaknesses:**

- Sometimes the authors make strong claims that are not necessarily justified. For example, on page 5, the authors claim that, "For fair data deals, it is necessary to understand when, where, and how data is used in the AI pipeline." It's not clear to me why this tracking is necessary or how fairness can be imposed (or to what notion of fairness).
- It's not clear to me whether EDVEX is a new framework or one that already exists in the literature. *If EDVEX is completely brand new, the position track is not suitable for this paper as it should undergo a thorough peer review with more evidence of its possible success.*
- There are a lot of open directions and problems and little discussion on the possible downsides that could arise from this. While the authors do acknowledge that malicious actors could "game" the data valuation system (page 7), it still feels like there should be a stronger discussion on possible pitfalls

**Questions:**

- Is EDVEX a new framework?
- There are a lot of open questions. What should be prioritized?
- The authors mention that there is a challenge with tracking provenance and dealing with data privacy regulation. How do the authors suppose someone could manage these?

**Alternative Position:**

Yes, and alternative positions are well-considered and addressed by the argument

**Author Identification:**

No.

**Context:**

2

**Discussion:**

4

**Ethics:**

["NO or VERY MINOR ethics concerns only"]

**Position:**

Yes, the paper argues for or against a position related to machine learning.

**Support:**

3

**Thoroughness:**

3

---

### Official Review · Reviewer_nFi7 · 2025-08-13

**Significance:** 3
**Presentation:** 3
**Rating:** 4
**Confidence:** 2

**Summary:**

The paper argues that the current machine learning (ML) value chain is economically unsustainable due to an "economic data processing inequality" that strips value from data generators while enriching aggregators and model monetizers. By analyzing 51 public data deals, the authors identify three structural flaws that systematically disadvantage data originators. They propose the Equitable Data-Value Exchange (EDVEX) framework to create fairer, more transparent data markets that align incentives and improve long-term sustainability for all participants.

**Strengths:**

The authors present a clear conceptual framework (EDVEX) grounded in empirical evidence from 51 publicly disclosed data deals. Their analysis articulates structural economic problems in the ML data economy with clarity, combining economic theory with practical examples. The integration of technical, economic, and policy considerations makes the proposal relevant across multiple disciplines. Visual diagrams and well-structured sections enhance accessibility and engagement for both technical and non-technical readers.

**Weaknesses:**

The empirical dataset is limited to publicly disclosed deals, potentially omitting the most impactful or representative transactions. The framework, while conceptually robust, is still theoretical and lacks a concrete implementation or pilot study to validate feasibility. Some recommendations, such as global provenance standards, face substantial political, legal, and logistical barriers that are underexplored. The scope is broad, which can dilute the depth of analysis on critical technical aspects like utility-driven pricing algorithms.

**Questions:**

1. How might EDVEX be adapted for sensitive data domains (e.g., healthcare, defense) where provenance tracking and open pricing could be restricted by law?
2. What governance or incentive structures could prevent powerful aggregators from capturing and dominating an EDVEX-like marketplace?
3. Could synthetic data generation alter the economic balance the authors describe, and how would EDVEX handle valuation of synthetic contributions?
4. What metrics or experiments could be used to validate whether EDVEX truly improves bargaining symmetry and economic equity in practice?

**Alternative Position:**

Yes, and alternative positions are well-considered and addressed by the argument

**Author Identification:**

No.

**Context:**

3

**Discussion:**

4

**Ethics:**

["NO or VERY MINOR ethics concerns only"]

**Position:**

Yes, the paper argues for or against a position related to machine learning.

**Support:**

3

**Thoroughness:**

3

---

### Note · Authors · 2025-08-28

**1-11 Submit Again:**

Probably yes

**1-1 Submission Process:**

3

**1-3 Future Development:**

1. Share exact review criteria before submission deadline with authors to reduce misunderstandings between reviewers and authors

2. Explain the review/rebuttal/survey/discussion process clearly beforehand

3. Implement reviewer training to ensure consistency with track guidelines

**1-4 Interest:**

["Panel discussions with other position paper authors", "Structured debates on controversial topics", "Workshops for developing position papers", "Mentorship programs for early-career researchers"]

**1-5 Thoughtful:**

6

**1-6 Supportive:**

6

**1-7 Technical Aspects Versus Position:**

5

**1-8 Gate Keeping:**

10

**1-9 Camera Ready Changes:**

1. Update framework Figure 5 as pure TikZ figure

2. Address reviewer feedback:
2.1 Extend section 3.2 (Incentives for Implementing EDVEX) to include considerations raised by Reviewers nFi7, nioU and FK6j regarding practical challenges of addressing the economic data processing inequality
2.2 Clarify the relationship between provenance and equity in Section 3.1 (Lineage Tracking and Auditable Provenance), as suggested by Reviewer nioU
2.3 Add discussion of implementation challenges and transition barriers as identified by Reviewer FK6j

**3-1 Review Response1:**

nFi7

**3-2 Reaction To Review1:**

While reviewer nFi7 acknowledges the strengths of our framework, their requests for implementation details and non-public information are in our eyes unactionable and irreconcilable with the track requirements ("no technical papers"). The disconnect between their their positive scores ('Support: 3: good, Significance: 3: good, Presentation: 3: good, Context: 3: good, Discussion: 4: very likely, Alternative Position: Yes') and their final rating ("Borderline reject") is inconsistent. In our eyes, this review requires third-party adjudication to ensure fair evaluation of our submission. Regarding the individual points raised:

1. Sensitive Domains: EDVEX's modular design enables domain-specific adaptations. We explicitly identify privacy-preserving utility estimation as an open research problem, recognizing regulatory constraints while proposing technical directions.

2. Preventing Aggregator Capture: This question validates our analysis of current power asymmetries and precisely motivates the proposed framework. EDVEX's design specifically counters capture through: (1) Dynamic data unions that prevent long-term lock-in, (2) Utility-based pricing that reduces negotiation leverage, (3) Auditable provenance that enables direct creator-consumer relationships. Network effects can work for smaller players when they can efficiently organize around high-value, task-specific contributions.

3. Synthetic Data Impact: This question has been addressed comprehensively in §5. Synthetic data creation requires substantial resources, making it a form of data contribution within EDVEX. Model collapse research confirms real-world data remains essential for long-tail domains.

4. Validation Metrics: This question is beyond the scope of a position paper. Position papers propose frameworks; specific evaluation methods depend on implementation context and application domains—exactly the research directions we're establishing.

**3-3 Review Response2:**

nioU

**3-4 Reaction To Review2:**

Reviewer nioU was very positive about our work, describing it as "eye-opening" and "thoroughly informative." Their enthusiasm for the public data analysis was particularly gratifying. The reviewer appreciated that we "successfully acknowledge opposing viewpoints without dismissing them outrightly". Overall, this review was very supportive while still providing constructive, actionable feedback:

1. Provenance Tracking Necessity: The reviewer questions our claim about provenance tracking being necessary for fair data deals. This follows directly from our utility-based compensation premise: to pay data generators proportional to contribution, we must (1) identify who provided data, (2) measure contributions to model performance, and (3) track usage patterns. Without provenance, the system defaults to zero compensation or utility-blind buyouts—both demonstrably unfair per our empirical analysis. Our fairness notion is utility-based: data value should reflect its contribution to ML tasks, implemented through utility-driven pricing.

2. EDVEX Novelty and Track Appropriateness: EDVEX is a new conceptual framework synthesizing existing components (data cooperatives, provenance tracking, utility pricing) into an integrated solution addressing all three structural faults simultaneously. This is precisely appropriate for the position track. We're proposing a research agenda and conceptual framework to guide community efforts, not claiming technical completeness requiring full peer review.

3. Limited Pitfalls Discussion: This feedback is fair. While we address some implementation challenges (technical primitives in §3.1, market dynamics in §5), further treatment of downsides would strengthen our argument. However, comprehensive pitfall analysis exceeds position paper scope—our goal is establishing research frameworks rather than exhaustive risk assessment.  The reviewer's acceptance rating suggests this limitation doesn't undermine our core contribution.

**3-5 Review Response3:**

FK6j

**3-6 Reaction To Review3:**

Reviewer FK6j highlighted that our paper "excels in combining rigorous empirical analysis with comprehensive theoretical framework development" and noted that our analysis of 51 real data deals "provides concrete evidence for abstract economic arguments." Their recognition that we "successfully bridge multiple disciplines" and make contributions to understanding both ML systems and economic foundations was particularly gratifying.

We note that most of the reviewer's concerns stem from overlooking existing sections (e.g., Section 3.2 on incentives) or expecting implementation details beyond a position paper's scope of establishing research frameworks. Addressing the main concerns:

1. Shapley Value Scalability: The reviewer correctly identifies computational challenges but overlooks that we already cite recent breakthroughs addressing this exact issue. "Data Shapley in One Training Run" (Wang et al., ref 50) demonstrates scalability to pre-training scale, making this approach practically feasible for EDVEX implementations.

2. Adoption Incentives: Section 3.2 explicitly addresses this concern with detailed analysis of stakeholder incentives. We demonstrate compelling motivations for each actor: data contributors gain market access and fair compensation, AI developers get better data discovery and reduced legal risk (citing Meta's $1.4B+ settlements), and platform providers capture new market opportunities. The reviewer may have missed this comprehensive treatment.

3. Technical Complexity: This concern actually validates our core contribution. Position papers exist precisely to tackle complex, multi-faceted problems that require coordinated community effort. Rather than being a weakness, the technical complexity justifies our systematic identification of open problems and research agenda. No single paper could implement the full framework—our goal is establishing the conceptual foundation for distributed research efforts.

---

### Meta-Review · Area_Chair_FPbf · 2025-09-12

**Rating:** 7
**Confidence:** 4

**Strengths:**

This position paper provides contribution on the economics of machine learning data markets. It clearly frames the problem of economic data processing inequality and supports the position with the analysis of 51 public data deals. The authors discuss a new conceptual framework, EDVEX, that brings together ideas from economics, machine learning, and policy. The paper is accessible and easy to follow.

**Weaknesses:**

The proposed framework is largely theoretical, without concrete pilots or validation studies (which is acceptable in a position paper). The use of Shapley values for revenue sharing, or sandbox evaluation protocols for utility estimation, may not scale to millions of contributors and datasets.

**Questions:**

Could synthetic data change the economic balance described, and how should EDVEX account for synthetic contributions?

What should be prioritized among the many open problems the framework identifies?

How EDVEX would coexist with, or be constrained by, current laws and regulations (GDPR and the EU Data Act)?

**Ethics:**

There are no ethical concerns.

**Thoroughness:**

2

---

### Decision · Program_Chairs · 2025-09-26

Accept